# Blast Hole Pressure Measurement and a Full-Scale Blasting Experiment in Hard Rock Quarry Mine Using Shock-Reactive Stemming Materials

**Younghun Ko** [1,*] **, Chanhwi Shin** [2] **, Youngjun Jeong** [3] **and Sangho Cho** [2,4,*]

1    Department of Geotechnical Engineering Research, Korea Institute of Civil Engineering and Building Technology, 283 Goyang-daero, Ilsanseo-gu, Goyang-si 10223, Korea
2    Department of Energy Storage & Conversion Engineering of Graduate School, Jeonbuk National University, 567 Baekje-Daero, Deokjin-gu, Jeonju-si 54896, Korea
3    Seokseong Blasting Construction Co., Ltd., Wolgye-ro 37-gil, Gangbuk-gu, Seoul 01226, Korea
4    Department of Mineral Resources and Energy Engineering, Jeonbuk National University, 567 Baekje-Daero, Deokjin-gu, Jeonju-si 54896, Korea
*    Correspondence: younghunko@kict.re.kr (Y.K.); chosh@jbnu.ac.kr (S.C.)

**Abstract:** By increasing the effectiveness of the energy generated by the explosive charge inserted into a blast hole, stemming increases rock fragmentation. Missing or improper stemming, which can lead to the detonation gas escaping from the blast hole in advance, results not only in the waste of explosive energy and poor fragmentation but also in environmental problems, such as ground vibration, noise, flying rocks, back breaks, and air blasts. In this study, a stemming material based on a shear thickening fluid (STF) that reacts to dynamic pressure was developed. Two blasting experiments were conducted to verify the performance of the STF-based stemming material. In the first experiment, the pressure inside the blast hole was directly measured based on the application of the stemming material. In the second experiment, full-scale bench blasting was performed, and the blasting results of sand stemming and the STF-based stemming cases were compared. The measurement results of the pressure in the blast hole showed that when the STF-based stemming material was applied, the pressure at the top of the blast hole was lower than in the sand stemming case, and the stemming ejection was also lower. Full-scale bench blasting was conducted to compare the two types of stemming materials by evaluating the size of the rock fragments using image processing. The results of the two blasting experiments helped to verify that the blockage performance of the STF-based stemming material in the blast hole was superior to that of the sand stemming material.

**Keywords:** blasting experiment; stemming material; shear thickening fluid; sand; blockage performance

## 1. Introduction

Stemming is a process applied to blast holes to prevent gases from escaping during detonation. A stemming material helps confine the explosive energy for a longer duration. Without stemming, up to 50% of the explosive energy can escape through the borehole [1]. Proper stemming has been shown to improve explosive efficiency by over 41% [2]. Further, employing even the least efficient stemming materials can boost the usable energy of an explosion by 60%, while the most efficient stemming materials can increase it by up to 93% [3].

Missing or improper stemming, which can lead to the detonation gas escaping from the blast hole in advance, results not only in the waste of explosive energy and poor rock fragmentation but also in environmental problems, such as ground vibration, noise, flying rocks, back breaks, and air blasts [4].

Smaller amounts of explosives may be used to produce the same blasting effects if explosive energy was used more effectively [5,6]. Improvements in fragmentation will

result in lower second breaking work costs. Proper stemming can reduce costs and improve the productivity and profitability of a mining operation. The main objective of rock-blasting is fragmentation by explosive. The loading and hauling operations of a mining operation, particularly the crushing line, profit greatly from good fragmentation [7]. Additionally, cracks are generated over a larger area in the rock mass using proper stemming. These cracks propagate, interconnect, and cut the rock mass; thus, the block size and distribution after blasting satisfy the construction or mining requirements. Therefore, reasonably selecting the stemming material is of particular importance for improving the blasting effect, increasing the efficiency of explosives, and obtaining ideal blasting fragmentation.

In the mining industry, blast holes are sealed with three different sorts of stemming materials: colloidal, liquid, and solid. Additional research on the performance of stemming materials is required. Li et al. [8] used a water stemming technique that involves putting water-filled polyvinyl plastic bags within blast holes. A water-silt composite-stemmed blasting method for tunnels was proposed [9] to increase rock breakage, reduce dust, and use fewer explosives.

In most cases, using fluid-type stemming inside a blast hole as a stemming material produced good results. Fluid has a higher density than air, and even at extremely high pressures, the compression of water is significantly lower than that of air [10].

Under dynamic loading, Zhu et al. [11] performed an AUTODYN numerical analysis using a variety of stemming materials, such as fluid (water), sand, and air, placed in the area between the internal explosives and the hollow wall inside the blast hole. The water stemming case, which was also the best medium for shockwave transmission, produced the largest fracture area. Additionally, the presence of fluid (water), which results in the deformation and displacement of the rock, causes the shock wave to reflect and bubble pulse, which contributes to the high stress exerted during this process.

The most effective method to evaluate the stemming effect is to conduct field experiments. A stemming performance test of a small-scale model was developed, and the results showed that different stemming materials have different functionalities, which can significantly influence the efficiency of rock breaking [12].

Kopp [13] suggested a simple physical model for predicting the time required to eject stemming. This model depends only on the inertia of the stemming material. The frictional forces that resist the movement were omitted. The stemming performance of the stemming material can be evaluated using the initial ejection velocity of the stemming part at the entrance of the blast hole [14].

The momentum of the stemming structure based on the explosive load can be calculated using the mass of the stemming structure and the initial velocity of the stemming part when the stemming is ejected into the orifice.

As a highly capable method in the mining industry, image analysis techniques have been used to predict rock fragmentation by blasting. These techniques are capable of visual processing, thereby serving as an appropriate alternative to low-accuracy methods [15]. Over the past few decades, various image analysis software packages, such as Split-Online, Split-desktop, Gold-Size, and Wip-Frag, have been developed, and their applications in the mining industry and mineral processing have been reported. The main advantages of these software packages are their integration and lack of disruption [16].

The specific charge is mainly used as an indicator to predict the blasting effect, but the amount of powder used per unit volume of crushed rock cannot properly reflect the influence of energy change in the blasting hole; therefore, the pressure in the blasting hole must be estimated and used to understand this [17]. However, the blasting pressure has relied on calculations rather than direct measurement. Recently, the blasting pressure has been estimated through a numerical analysis approach, but it is difficult to predict the explosion reaction of explosives acting on rocks, an anisotropic material. Therefore, the concepts of abnormal explosion and ideal sealing are used to calculate the pressure in the blast hole, assuming that there is no external influence [18]. Therefore, field experiments are being conducted to directly measure the pressure in the blast hole. However, the pressure

probe used is very expensive, and the sensor is only used once because of the extreme conditions generated by the blast, thereby requiring a significant financial investment. Therefore, in most cases, a pressure measurement sensor is inserted into the dummy hole, and the blasting pressure is indirectly measured based on the impact pressure propagated through the rock [19].

In recent years, a new intelligent material named shear thickening fluid (STF) has been widely used in energy absorption research [20–25]. STF exhibits an intense viscosity jump under shock load; as a result, it has been used in various applications, such as liquid body armor [26]. However, limited research on the application of STF for industrial blasting or as a stemming material is currently available.

In this study, a shock-reactive stemming material was developed that behaves similar to water in terms of shockwave propagation and has a high shear strength for dynamic shocks. The STF is characterized by its reversible energy absorption behavior under impulse loading. Its remarkable energy absorption capacity is attributed to viscous dissipation during shear and compression thickening. The STF-based stemming material was developed based on the following advantages. (1) STF has excellent sealing properties as it is a fluid-based material. (2) Its viscosity rapidly changes because of external shock, while material compaction or deformation is minimal with respect to the dynamic gas pressure in the blast hole. (3) Using starch as the main base material reduces costs.

Two blasting experiments were conducted to compare and contrast the blast effects of the developed stemming material and those of commonly used blasting stemming materials. The first is an experiment in which the blast hole pressure and stemming ejection are directly measured, and the second is an experiment to verify the stemming performance by analyzing the assessment of rock fragmentation through a full-scale blasting experiment.

## 2. Materials and Methods

### 2.1. STF-Based Stemming Material Rheology Tests

A dense colloidal dispersion of solid nanoparticles in a carrier fluid is known as an STF [27]. When a shear force is applied, the random distribution of particles in the dispersion initially emerges in an ordered fashion because the hydrodynamic forces are greater than the repulsive forces operating between the interstitial gaps the particles have generated. The order–disorder theory, put forth by Hoffmann in 1972 [28], is represented by this arrangement of particles. Large hydrodynamic forces tend to push out the fluid between the interstitial spaces with rising shear rates, leading to the production of hydroclusters. The hydroclustering mechanism proposed by Brady and Bossis in 1985 is comprised of this phenomenon [29]. These clusters are stress-bearing elements that lead to particle jamming, when additional shearing pressures are applied.

STFs behave by increasing the dynamic viscosity under the application of shear stress. When tightly packed particles combine with enough liquid to cover the spaces between the particles, dilatancy occurs. The fluid serves as a lubricant at low speeds, facilitating easy movement of the dilatant substance.

Because of the increased friction caused by the inability of the liquid to fill the gaps left by the particles at greater velocities, the viscosity also increases. The STF is also non-Newtonian in nature because its viscosity depends on the shear rate or shear rate history. This behavior is a type of deviation from Newton's law and is controlled by factors, such as particle size, shape, and distribution. Empirical studies have also shown that shear thickening effects vary with different particles and additive concentrations, as well as with the molecular chain of the additives [30].

Shear thickening is a reversible phenomenon governed by a power law model. Generally, a non-Newtonian fluid is described using the power law model expressed in Equations (1) and (2).

$$\tau = K\left(\frac{\partial \mu}{\partial y}\right)^n = K(\gamma)^n = \tau = K(\gamma)^{n-1}(\gamma)^1 \ , \ \tau = \mu_{apparent}(\gamma), \tag{1}$$

$$\mu_{apparent} = K(\gamma)^{n-1} \tag{2}$$

where $\tau$ is the shear stress exerted by the fluid, $K$ is the fluid viscosity, $\mu$ is the shear deformation, $y$ is the distance from the reference layer, $\frac{\partial \mu}{\partial y}$ is the strain rate, $n$ is the flow behavior index, and $\mu_{apparent}$ is the apparent viscosity.

As shown in Figure 1, the fluid behaves as a Newtonian fluid at $n = 1$ and exhibits shear thinning properties when $0 < n < 1$. Moreover, several dispersions and liquid polymers exhibit shear thinning behavior for $n$ values between 0.3 and 0.7. However, this depends on the particle concentration and molecular weight of the carrier fluid.

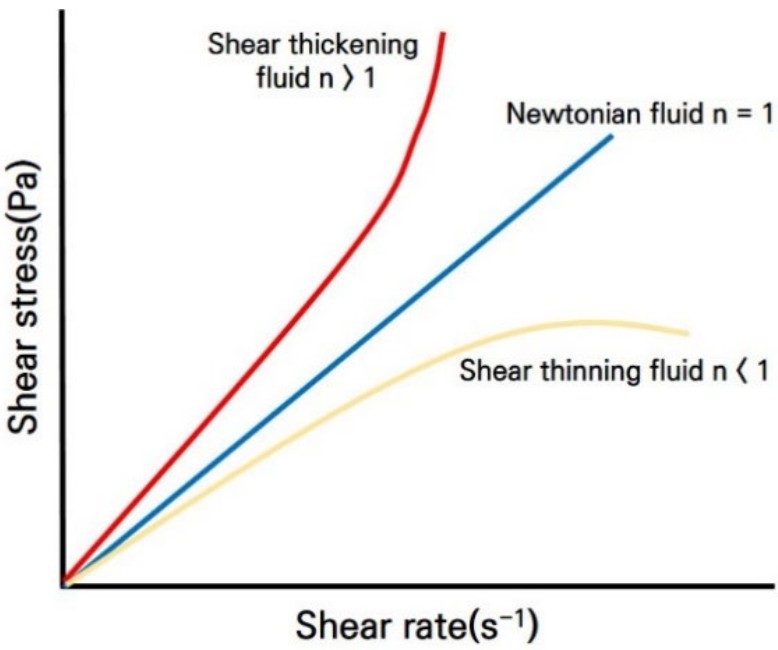

**Figure 1.** The shear thickening process of a shear thickening fluid.

The shear thickening effect of STF is demonstrated by the lower rate of increase in the shear stress in the low-shear-strain-rate regions and a higher rate of increase in the high-shear-strain-rate regions (Figure 1). Figure 1 also depicts the relationship between the shear stress and shear strain rate of the STF. The potential of a STF to improve the effectiveness of body armor against ballistic impacts and stab resistance has been thoroughly researched [31,32]. Further research is necessary, nevertheless, to fully understand the potential impact of STF on the stemming of blast holes. The motivation of this study is to harness the strength of the STF through flexible deployment and relatively easy stemming, which can help dissipate pressure wave loading around the rock mass during an explosion. The rheological behavior of the non-Newtonian fluid was measured using a rheometer.

In this study, the STF-based stemming material was mainly based on corn starch, while xanthan gum and guar gum were blended to increase the viscosity. Sodium benzoate was used to prevent the putrefaction of starch, and nitroglycol and salt were added to prevent freezing in winter.

The STF samples were sandwiched between a cone plate and the foundation support of a rheometer (Anton-Paar MCR301 rheometer) for rheological experiments. During the studies, the shear rate applied to the sample was increased from 0 to 100 s$^{-1}$, and all of the tests were carried out at a temperature of 25 °C.

A schematic of the rheometer and the results of the rheological tests conducted on the STF-based stemming material are shown in Figures 2 and 3. The rheology tests were performed on the STF samples of 30, 45, and 55 wt.% corn starch suspensions. In the case of the 30 wt.% corn starch suspension, no significant shear thickening is observed. However, as the starch content increases, the particle content exceeds the ratio of the dispersion

medium; therefore, the distance between the corn starch particles decreases, and the shear thickening effect increases. For the sample made up of 55 wt.%, a shear thickening effect is attained at a critical shear rate of 85 s$^{-1}$. The STF initially experiences marginal shear thinning, which then grows with the shear rate. In particular, the viscosity of the STF suddenly increases as the shear rate reaches a critical value, indicating a shear thickening phenomenon. However, the viscosity of the STFs sharply decreases after a period of shear thickening. The critical shear rate of the STF sample is approximately 85 s$^{-1}$, and the maximum viscosity of the STF samples is 543 Pa.

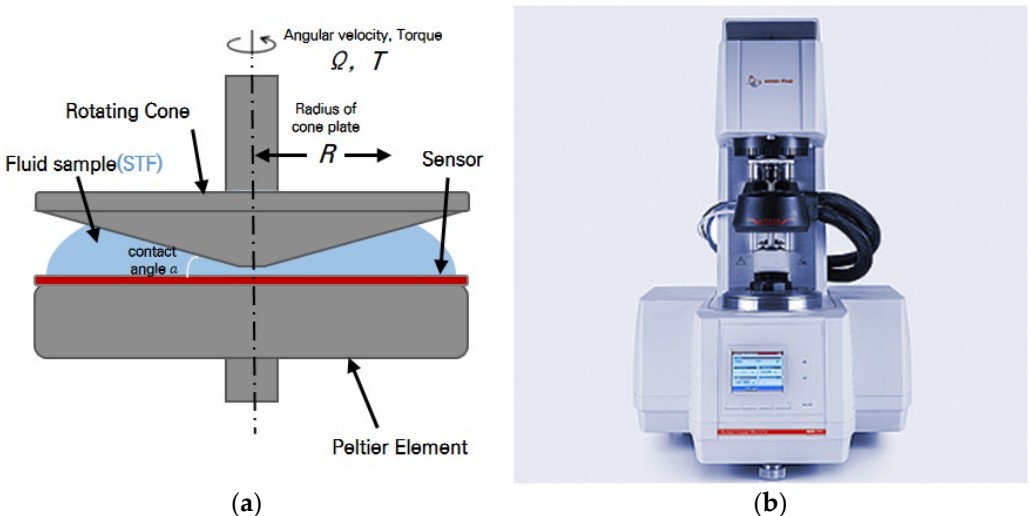

(**a**)                (**b**)

**Figure 2.** (**a**) Rheometer schematic and (**b**) Anton-Paar MCR301 rheometer.

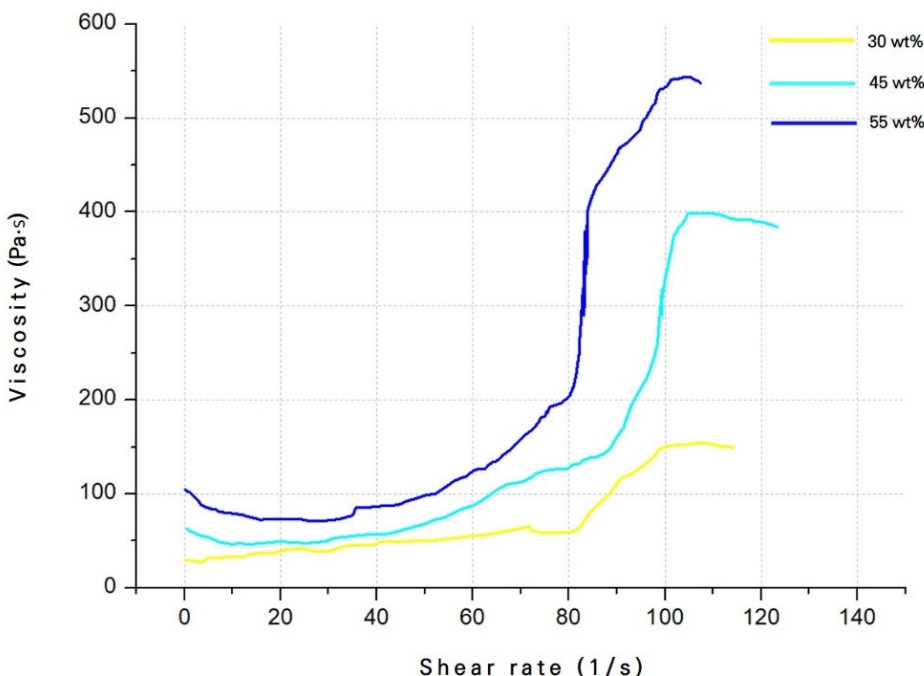

**Figure 3.** Results of the shear thickening fluid based stemming material's rheological tests.

The STF-based stemming material image and package products are shown in Figure 4. The hammer recoils upon impact with the STF suspension surface, similar to hitting a true solid interface. It is a suspension of starch powder with a diameter range of 5–20 μm in water. The STF was created using a mixture of mechanical and ultrasonic mixing at a concentration of 55 wt.%. This weight percentage was chosen to preserve a viable solution,

while ensuring the shear thickening tendency [33]. According to previous experimental results [34] for 52.5–55 wt.% corn starch, after the impact of rock falling on the suspension surface, the rock recoiled, similar to hitting a true solid interface.

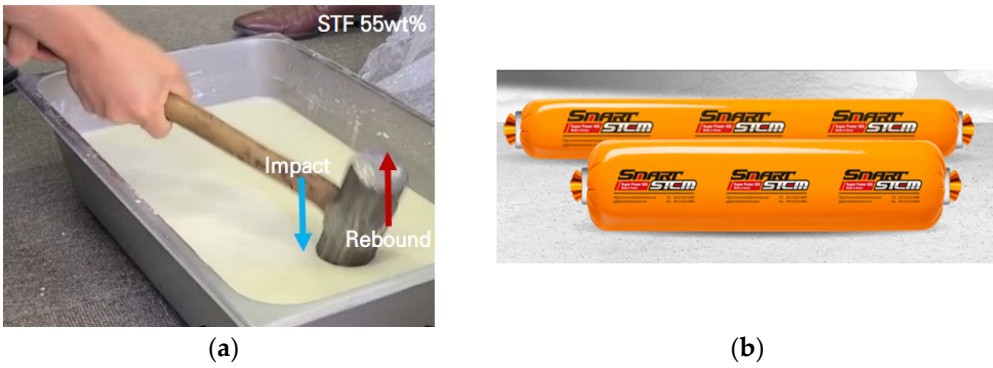

| (**a**) | (**b**) |

**Figure 4.** (**a**) Shear thickening fluid-based stemming material image and (**b**) package products.

### 2.2. Overview of the Blast Hole Pressure Measurements

In blasting, stemming constrains the blasting pressure in the blasting hole, leads to crack propagation through the behavior inside the blasting hole under the explosive pressure, and ultimately plays an important role in crushing the rock. Therefore, the behavior and control of the explosive pressure acting on the blast hole are important for effective blasting.

Laboratory- and field-scale experiments were conducted to measure the pressure inside the blast hole. However, because the sensor that measures the pressure inside the blast hole is expensive, there is a risk of failure owing to a strong impact. Therefore, the explosive pressure is measured using a one-time polyvinylidene fluoride (PVDF) sensor [35] or by drilling a dummy hole [36].

In this study, instead of using a one-time sensor or drilling a dummy hole to measure the pressure inside the blast hole, nylon tubes and water pressure measurement sensors that can be used multiple times were employed. Thus, the pressure generated by blasting was measured in the blasting hole without sensor damage or additional drilling.

### 2.3. Assessment for Rock Fragmentation of Bench Blasting

In this study, full-scale bench blasting was performed for each stemming material (sand or STF), and an image-based blast fragmentation method was applied to compare and evaluate the performance of each stemming material.

Sieving or screening is a direct and accurate method for evaluating the size distribution of particles or their fragmentation. This method is feasible for small-scale blasts or operations; however, it is costly and time-consuming. Rock fragments are screened through sieves of different mesh numbers for different fragment sizes, and the screened fragments are grouped based on their size. The nature of the blast was predicted by counting the number of fragments of each size [37].

WipFrag is an image analysis system for sizing materials, such as blasted or crushed rocks [38]. It has also been used to measure other materials, such as ammonium nitrate prills, glass beads, and zinc concentrates. WipFrag accepts images from a variety of sources, such as roving camcorders, fixed cameras, images, or digital files. It uses automatic algorithms to identify individual blocks and to create an outline "net" using state-of-the-art edge detection. If desired or necessary, manual intervention (editing the image net) can be performed to improve the fidelity. WipFrag measures a 2D net and reconstructs a 3D distribution using the principles of geometric probability. WipFrag supports two methods: Rossin Rammler and Swebrec. Two parameters were used by Rammler as key performance indicators (KPIs); $X_c$, known as the characteristic size of the distribution and more specifically D63.2, and $n$, the value of which is the measure of uniformity [39].

Analyzing every fragment in the rock muck pile is fortunately not necessary because it is widely accepted that the mass percentage of fragments smaller than any given size varies linearly with the fragment size when plotted in the Rosin–Rammler domain. By measuring only a sufficient number of particles, the slope and intercept of the Rosin–Rammler line can be confidently defined [40]. A Rosin–Rammler line can be expressed as in Equation (3).

$$R(X) = 1 - \exp\left[-\left(\frac{X}{X_c}\right)^n\right] \tag{3}$$

Here,

$R(X)$ = Cumulative fraction by weight undersize in relation to size $x$.
$X_c$ = Size modulus, which defines the characteristic size of the distribution.
$n$ = Distribution modulus, which defines the spread of the distribution.

For $R(X)$ = 0.5 (i.e., 50% of the fragments passing through the sieve), the value of $X_c$ can be measured as follows:

$$X_c = \frac{X_{50}}{0.693^{1/n}} \tag{4}$$

## 3. Blast Hole Pressure Measurement Experiment

### 3.1. Explosion Pressure Sensor Calibration

In this study, it was necessary to calibrate the pressure sensor to measure the explosive pressure inside the blast hole. Therefore, prior to this experiment, explosive pressure sensor calibration using water pressure was performed. The explosive pressure sensor was calibrated under the same installation conditions as those of the blasting site.

Calibration of the explosion pressure sensor was performed by filling the nylon tube connected to the explosion pressure sensor with water, connecting it with a water pump, and pressurizing the pressure port of the explosion pressure sensor under the conditions of 0 MPa, 25 MPa, and 50 Mpa three times each. The average and standard deviation of the results for three calibration tests was 0.984 ($\pm$0.0033) Voltage at 0 MPa, 2.98 ($\pm$0.0082) Voltage at 25 MPa, and 4.98 Voltage ($\pm$0.0144) at 50 MPa. In addition, Nonlinearity was 0.137% FS (Full Scale), and Accuracy was 0.86% FS. The results of the experiments performed are specified in Table 1, and the Voltage to pressure (MPa) correction equation is shown in Figure 5.

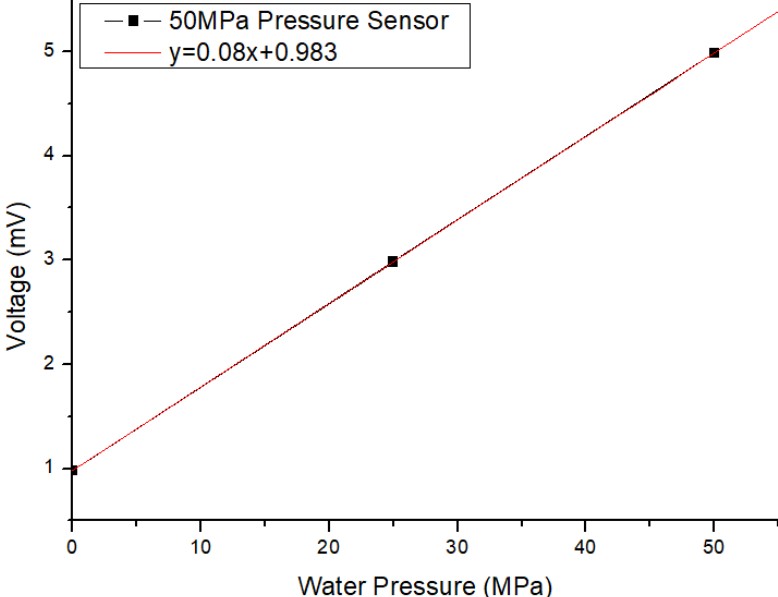

**Figure 5.** Explosive pressure bar calibration results.

**Table 1.** Results of the Explosion Pressure sensor calibration.

| No. | Pressure Range (MPa) | Analog Output (V) |
| --- | --- | --- |
| 1 | 0 | 0.985 |
| 2 | 25 | 2.98 |
| 3 | 50 | 4.984 |
| 4 | 0 | 0.988 |
| 5 | 25 | 2.97 |
| 6 | 50 | 4.955 |
| 7 | 0 | 0.98 |
| 8 | 25 | 2.99 |
| 9 | 50 | 4.987 |

*3.2. Blast Hole Pressure Measurement System*

The purpose of the experiment was to measure the explosion pressure in a blast hole for each string material and to evaluate the pressure resistance of the stemming material. Conventional sand and STF-based stemming materials were applied to evaluate their resistance capability under explosive pressure. Figure 6 shows the shape of each stemming material.

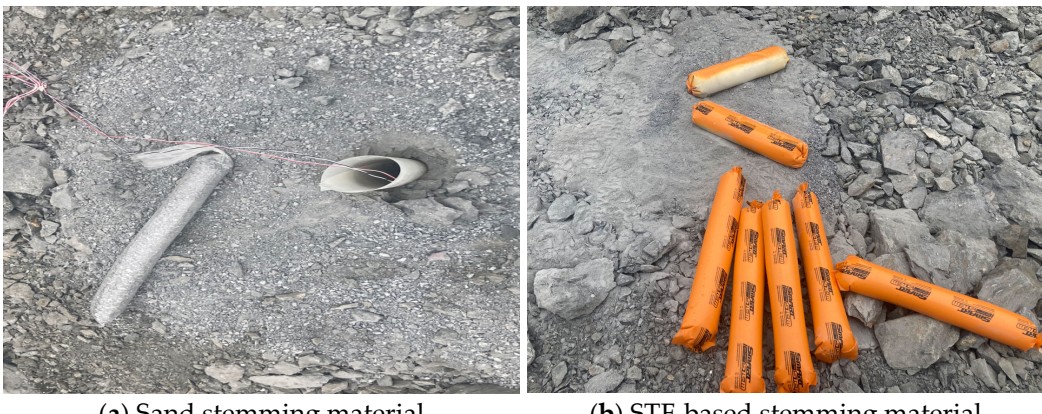

(**a**) Sand stemming material  (**b**) STF-based stemming material

**Figure 6.** Stemming materials used in blast hole pressure measurement experiment.

To measure the explosive pressure data from the blast hole, a pressure meter with cerabar (PMC) model of manufacturer Sensys capable of instantaneous dynamic shock pressure measurement was applied, and the MREL's MicroTrap was used to collect the data and set the detonation time trigger. The hydraulic shock pressure sensor had a pressure measurement range of 0–50 MPa. To measure the explosive pressure in the blast hole, the sensor was connected to a nylon tube filled with water using a coupling connector, placed in a water tube. To measure the explosive pressure in the blast hole, the sensor was connected to a nylon tube filled with water using a coupling connector, placed in a water tube. An explosive pressure-measuring device through hydraulic pressure was manufactured. Additionally, the manufactured water tube was protected with an industrial hose made of piezoelectric material to prevent damage when inserted into the blasting hole. Figure 7 shows the measuring tool applied to the blast pressure measuring system in the blast hole through hydraulic pressure. To collect the corresponding explosive pressure data as a time history at the same time as detonation using MicroTrap, a trigger line was attached to the explosion in the blast hole, and the break circuit trigger method was applied in which the connection signal was cut by the detonation of the explosive, and the measurement was finally started.

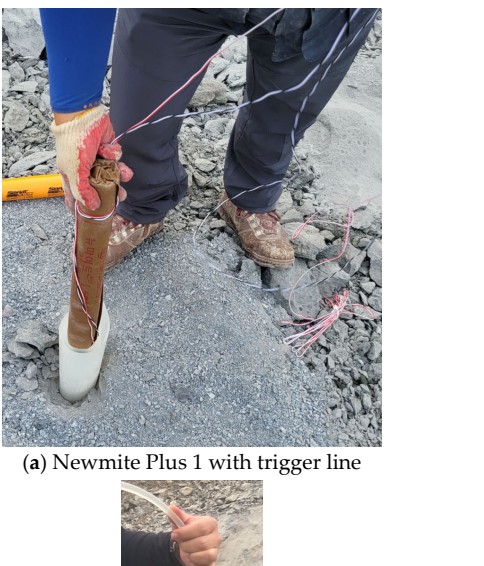

(**a**) Newmite Plus 1 with trigger line

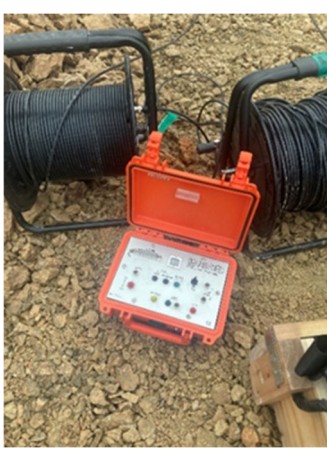

(**b**) MREL's MicroTrap

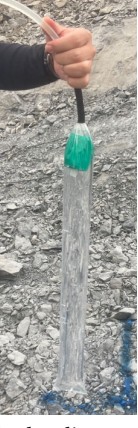

(**c**) Water tube for hydraulic pressure measurement

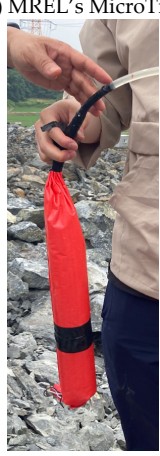

(**d**) Piezoelectric tube for water tube protection

**Figure 7.** Measuring tool applied to blast pressure measuring system.

The length of the blast holes drilled for measuring the impact pressure of the stemming material was 3.2 m, and the explosives charge length was 1 m. The emulsion series Newmite Plus 1 (Φ50 mm, 2.5 kg) manufactured by Hanhwa with an explosion speed of 5700 m/s were applied to the experiment. On top of the explosive, a test stemming material (sand or STF) of 0.6 m was applied; a 0.5 m water tube was inserted to measure the pressure caused by the explosion as the water pressure; and in the 1.1 m remaining at the top of the blast hole, general sand stemming was inserted. Figure 8 shows a schematic of the blast hole measurement system.

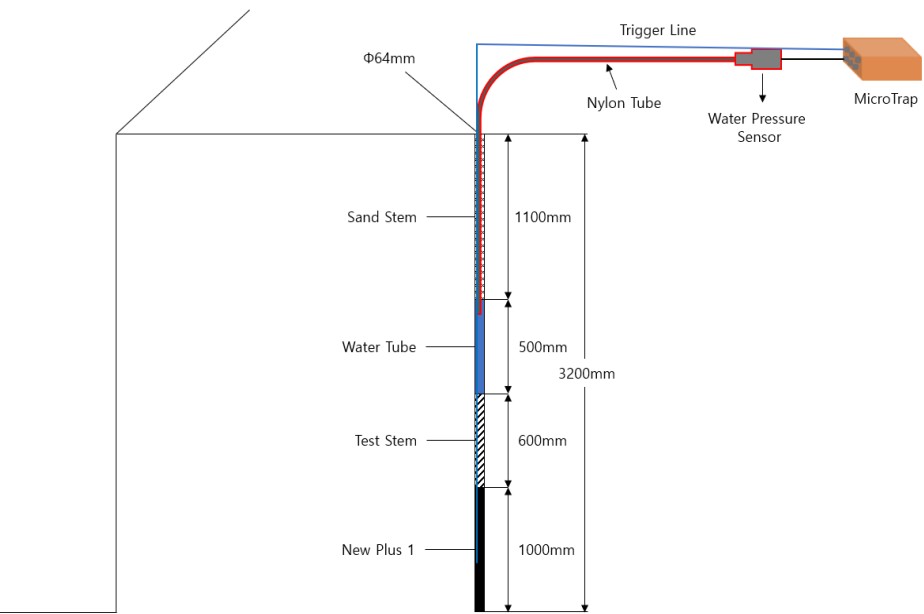

**Figure 8.** Schematic of the blast hole measurement system.

### 3.3. Experimental Results

In this experiment, a hydraulic explosive pressure propagation data measurement system was used to measure the pressure in the blast hole, and a comparative analysis was conducted on the blasting pressure behavior of the sand- and STF-based stemming materials. Table 2 presents the results of the blast hole pressure measurements.

**Table 2.** Result of the explosion pressure measurement in blast hole.

|  | Peak Pressure in Blasting Hole (MPa) | Explosive Pressure Arrival Time after Trigger (ms) | Explosive Pressure Duration in Blasting Hole (ms) |
| --- | --- | --- | --- |
| Sand Stemming | 5.84 | 12.24 | 16.82 |
| STF-based stemming | 2.80 | 31.82 | 21.81 |

The pressure in the blasting hole is 5.84 MPa for the sand stemming material and 2.80 MPa for the STF-based stemming material. Further, the blasting pressure by sand is two times higher than by STF-based stemming material. This is the pressure transferred to the water tube located above the test stemming material; therefore, a lower measured pressure value means that the loss of explosive pressure in the blast hole due to stemming transfer is lower.

The time taken from detonation to explosion pressure transfer is 12.24 ms in the sand stemming material and 31.82 ms in the STF-based stemming material; the measured value is significantly lower than that of the sand stemming material. This is the explosive pressure transfer time from the lower part of the blasting hole to the upper part of the blasting hole for the explosive detonation of the stemming material. Thus, the longer the measured explosive pressure delay time, the better the ejection resistance. In addition, the duration of the explosive pressure in the blast hole is 16.82 ms when using the sand stemming material and 21.81 ms when using the STF-based stemming material, an improvement of approximately 5 ms. This implies that the explosive pressure acted longer inside the blast hole, as long as the duration of the explosive pressure. Figure 9 shows the time-pressure hysteresis curve inside the blast hole.

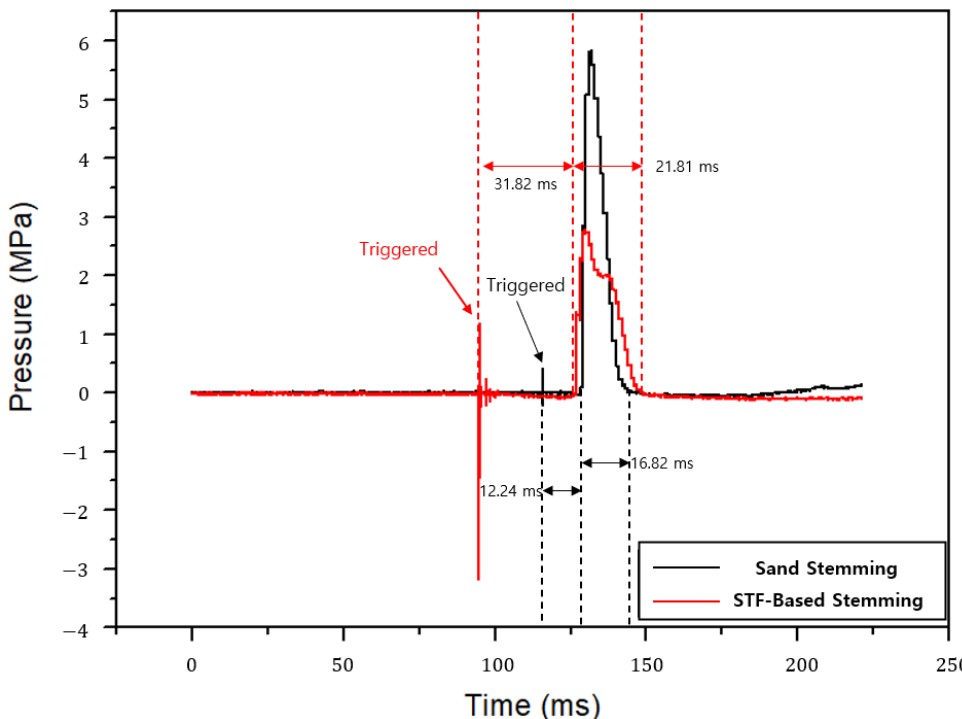

**Figure 9.** Time-pressure hysteresis curve inside the blast hole.

## 4. Full-Scale Blasting Experiment

### 4.1. Bench Blasting

The aggregate production step comprises blasting, a vibration feeder, a primary crusher, a secondary crusher, belt conveyors, and vibrating screens. The feed size of the jaw crusher operating in the A-aggregate mine requires a rock fragment size of at least 1000 mm or less. Therefore, to input the jaw crusher for primary crushing, it is necessary to perform secondary work on fragments by hydraulic rock breakers after blasting the rock, which incurs additional costs. Approximately $25,000 US dollars per month are required to operate large hydraulic rock breakers, and mine A has ten of such equipment. Figure 10 shows the bench blasting design applied to aggregate mine A, and the blasting results are compared by applying sand and STF-based stemming materials to each different bench part. Table 3 lists the main parameters of the full-scale bench blasting experiment. In the blasting experiment, the STF-based stemming was applied to the left part, and general sand stemming was applied to the right part.

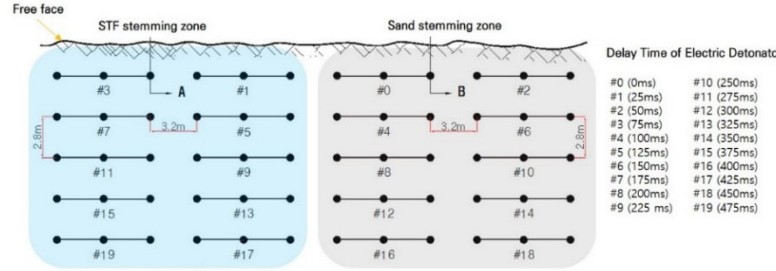

(**a**) Blast hole array and detonator delay time

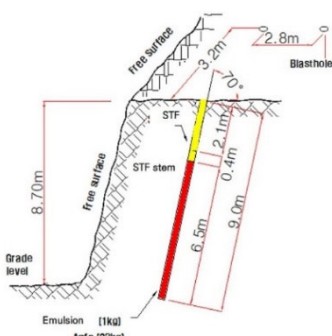

(**b**) Bench section (STF stemming)

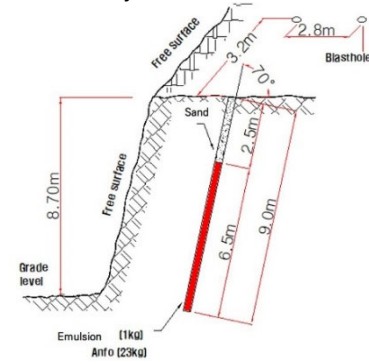

(**c**) Bench section (sand stemming)

**Figure 10.** Bench blasting design of full-scale blasting experiment.

**Table 3.** Main parameters of the full-scale bench blasting experiment.

| Parameter | Unit | Value |
|---|---|---|
| Hole diameter | mm | 75 |
| Hole length | m | 9.0 |
| Burden | m | 2.8 |
| Hole spacing | m | 3.2 |
| Charge per hole | kg/hole | 24.0 |
| Charge per delay | kg/delay | 72.0 |
| Charge type | - | Emulsion 1.0 kg ANFO 23.0 kg |
| Specific charge | $kg/m^3$ | 0.308 |
| Rock fracture per hole | $m^3$/hole | 77.95 |
| Number of holes | ea | 30 |
| Stemming length | M | 2.5 |
| Stemming type | - | STF or Sand |
| Total charge | kg | 720 |

The properties of the rock mass condition can have a significant influence on the fragmentation outcomes of the blast. Rock properties, such as compressive strength, porosity, Young's modulus, Poisson's ratio, and rock fracturing and jointing, can all influence fragmentation. In this experiment, by performing blasting at the same experimental area, the variation in results due to the rock mass condition difference was reduced as much as possible. The type of rock in this quarry mine is gneiss, and the uniaxial compressive strength is approximately 130–160 MPa. The rock density is approximately 2.6 $g/cm^3$, and the porosity is less than 0.15%. As shown in Figure 11a, a bench slope with a discontinuity direction and similar region was selected as the experimental site. The spacing of the discontinuities was observed to be approximately 1.2–1.5 m each, and they were under completely dry conditions. Figure 11b,c show the resulting image after the blasting. Under the same blasting conditions, the rock fragments in the STF-applied part are generally smaller than in the sand stemming part.

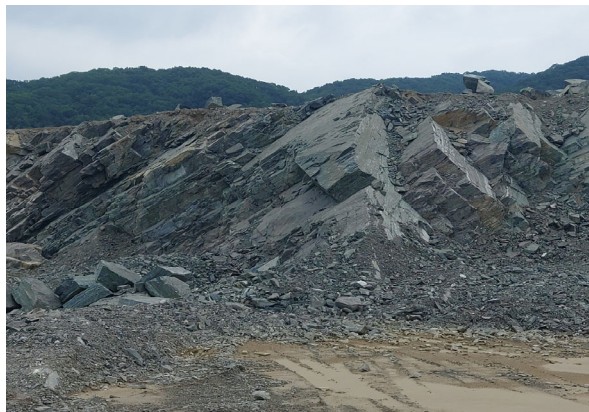

(**a**) Bench slope image of the experimental area

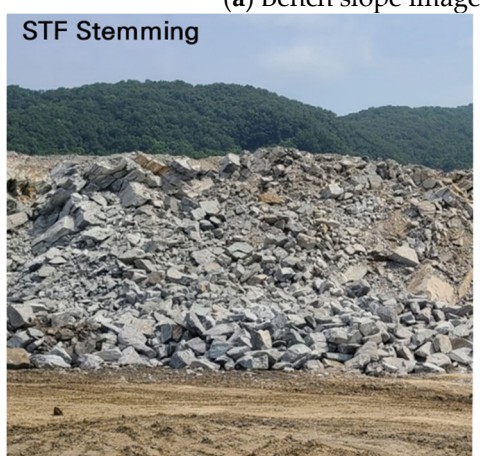

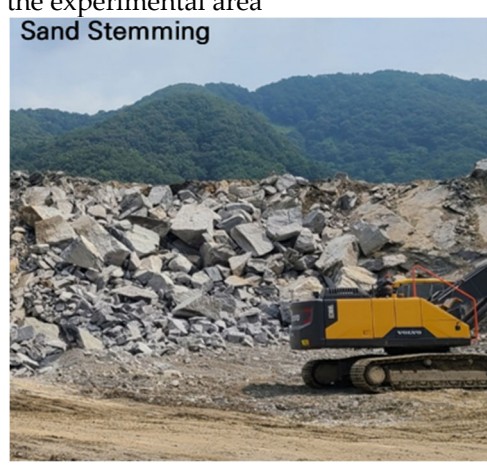

(**b**) Shear thickening fluid stemming

(**c**) Sand stemming

**Figure 11.** Bench blasting results obtained by applying different stemming materials.

### 4.2. Evaluation of Rock Fragmentation

Muck piles of fragmented rocks were photographed using a single camera from the top surface of the bench and the front. In the images, a reference scale was used for single-scale factor analysis by applying a square marker target of 18 cm in width and height. Table 4 shows the image-based sieving analysis results of the rock fragmentation of the muck pile after full-scale bench blasting. Figures 12–15 show the analysis of the rock fragment size distribution. Larger rock fragments are displayed in red in the resulting image.

**Table 4.** Results of rock fragmentation analysis.

| Location from Which the Image Was Captured | Stemming Type | Characteristic Size ($X_c$, mm) | Uniformity Index ($n$) | Average Fragment Size ($X_{50}$, mm) | Maximum Fragment ($X_{ma}$, mm) |
|---|---|---|---|---|---|
| Top of rock pile | Sand | 1149 | 1.66 | 836 | 1910 |
| | STF | 419 | 1.93 | 345 | 1.100 |
| Front of rock pile | Sand | 1234 | 2.07 | 1020 | 2610 |
| | STF | 938 | 2.01 | 772 | 2010 |

Images taken from the top of the bench muck pile are compared. In the bench region to which sand stemming was applied, the characteristic size ($X_c$) for evaluating the fragment size is approximately 2.7 times larger than that in the region to which STF-based stemming material was applied. Moreover, the uniformity index ($n$) for evaluating the particle size distribution of the fragment rocks of the muck pile is 1.66 for the sand stemming region and 1.93 for the STF-based stemming. A higher uniformity index indicates a more uniform distribution of the fragmented rock.

Similarly, images taken from the front of the bench muck pile are compared. In the bench region to which sand stemming was applied, the characteristic size ($X_c$) for evaluating the fragment size is approximately 1.3 times larger than that of the region to which the STF-based stemming material was applied; however, the uniformity index shows no significant difference between the two cases. This is expected to affect the uniformity index as a large rock fragment falls to the front of the muck pile after blasting. These large fragments fell from the top of the outermost bench and had relatively little effect on the explosive force. Moreover, it is for this reason that the average fragment size compared, respectively, at the front and top of the bench muck pile image shows a significant difference

The WipFrag 3 program automatically generates the histogram graph. The x-axis is a log graph showing the size distribution of the rock fragments. The y-axis is the rate of passing. Large fragments are marked with boxes in Figures 12–15. The size of the large fragments that must be subjected to secondary breaking with a breaker machine before being placed in the jaw crusher line is approximately 1000 mm or more in diameter. When STF is applied as a stemming, the average fragment size decreases, and the number of large fragments that require second breaking work is greatly reduced.

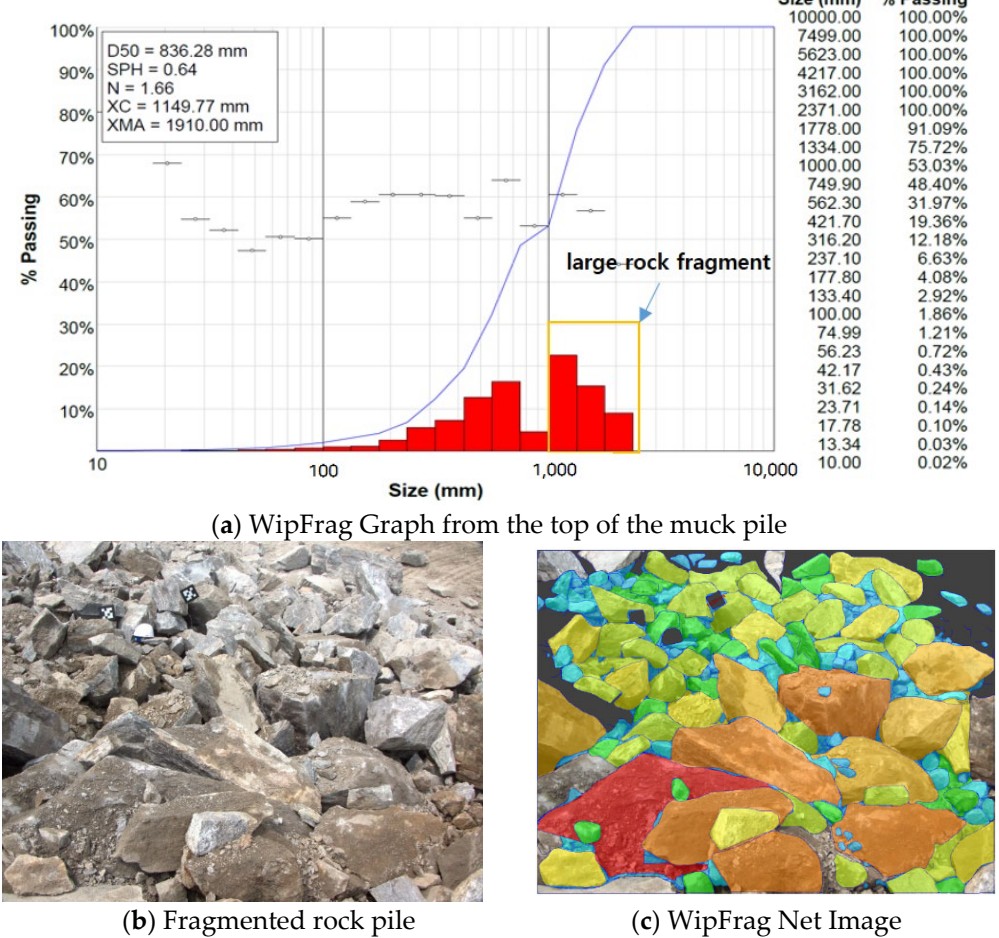

(**a**) WipFrag Graph from the top of the muck pile

(**b**) Fragmented rock pile          (**c**) WipFrag Net Image

**Figure 12.** Rock pile image, contouring, histogram, and cumulative size curve of the fragmented block (bench top image of sand stemming case after blasting).

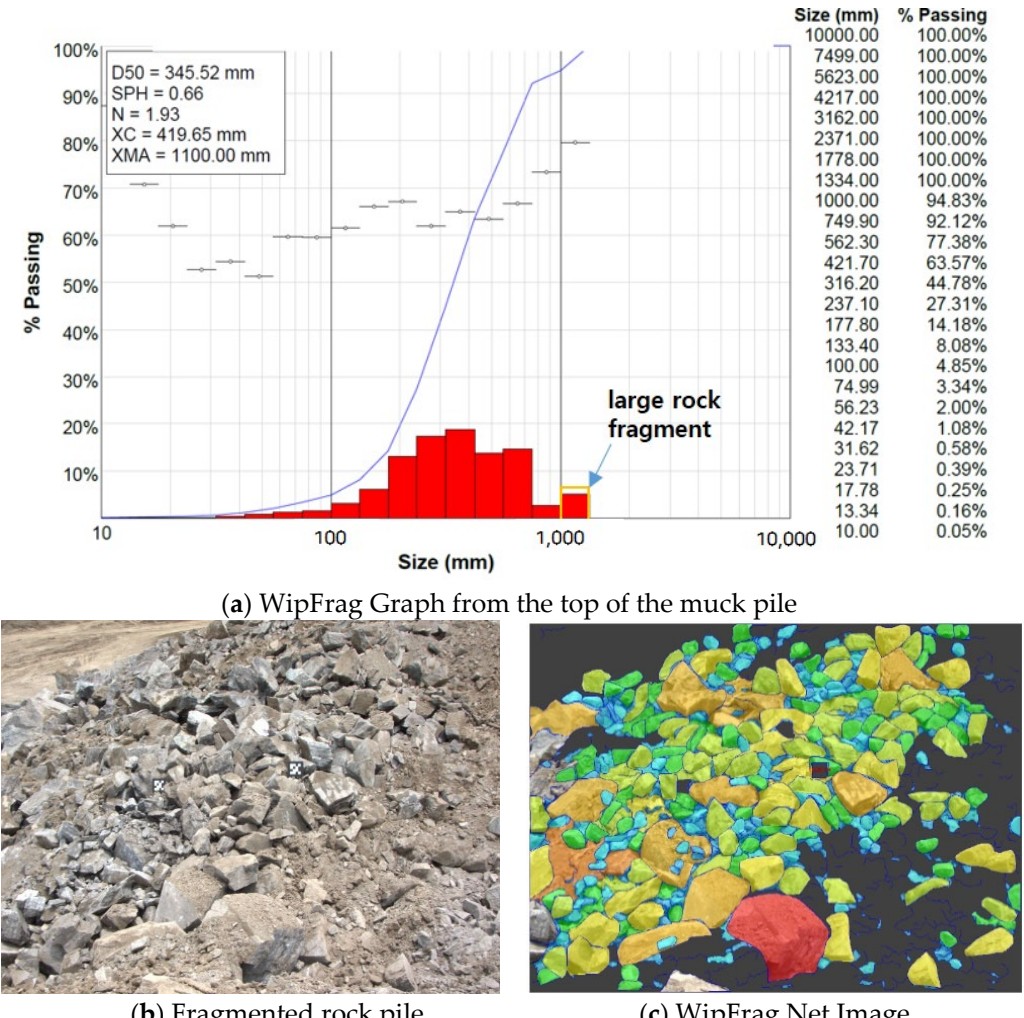

(**a**) WipFrag Graph from the top of the muck pile

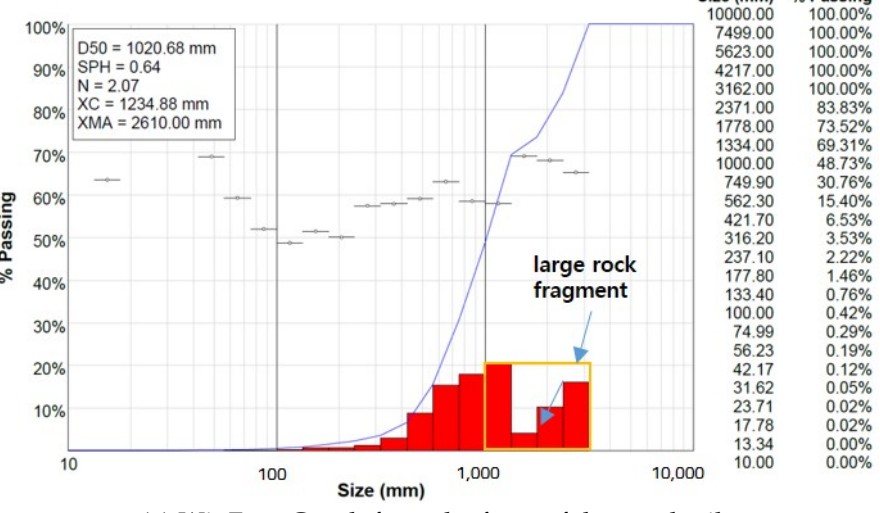

(**b**) Fragmented rock pile        (**c**) WipFrag Net Image

**Figure 13.** Rock pile image, contouring, histogram, and cumulative size curve of the fragmented block (bench top image of STF stemming case after blasting).

(**a**) WipFrag Graph from the front of the muck pile

**Figure 14.** *Cont.*

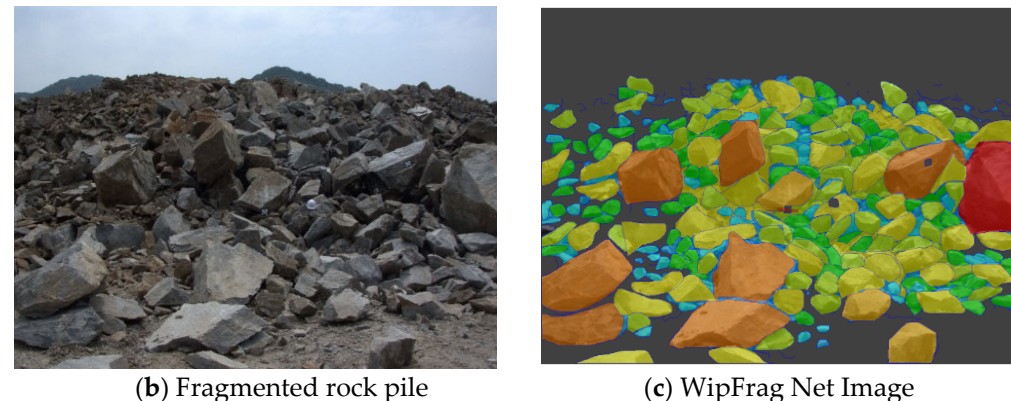

(**b**) Fragmented rock pile      (**c**) WipFrag Net Image

**Figure 14.** Rock pile image, contouring, histogram, and cumulative size curve of the fragmented block (bench front image of sand stemming case after blasting).

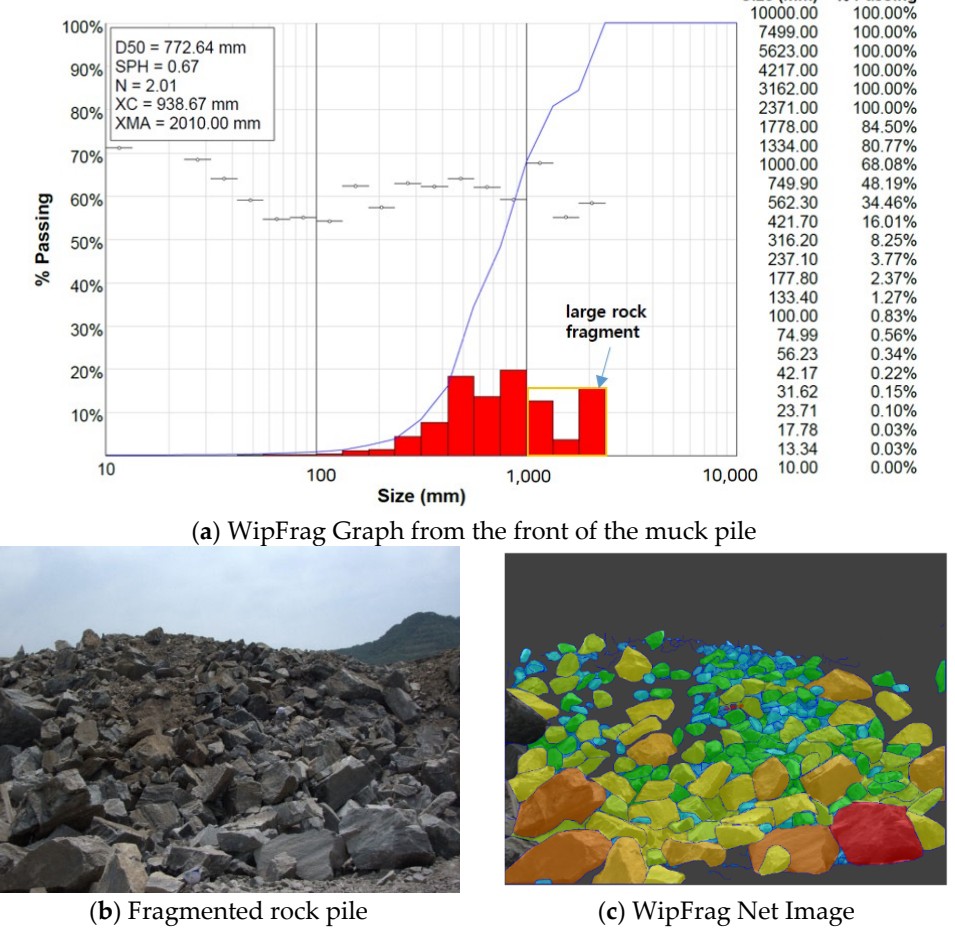

(**b**) Fragmented rock pile      (**c**) WipFrag Net Image

**Figure 15.** Rock pile image, contouring, histogram, and cumulative size curve of fragmented block (bench front image of STF stemming case after blasting).

## 5. Discussion

In this study, a material that instantaneously changes from shock load was developed as a blast stemming material and its performance was verified. Then, the pressure in the upper part of the stemming area was directly measured inside the blast hole. There are limited case studies that directly measure the pressure in the blast hole during bench blasting.

Previous research showed that stemming could increase the action time of the dynamic gas pressure in the blast hole and the efficiency of the explosives, reducing explosive consumption, as shown in Figure 16 [5,6]. Figure 16 depicts the time–pressure concept curves according to the stemming condition. In cases of missing or improper stemming, the pressure rapidly attenuates in the blast hole (Figure 16a), but proper stemming can increase the action time of the detonation gas inside the blast hole (Figure 16b). It is estimated that the shock pressure in the blast hole could be sustained for a longer time compared with that in the sand cases owing to the unique characteristics of STF. This is because STF is a smart-fluid type exhibiting an intense viscosity jump when subjected to loading.

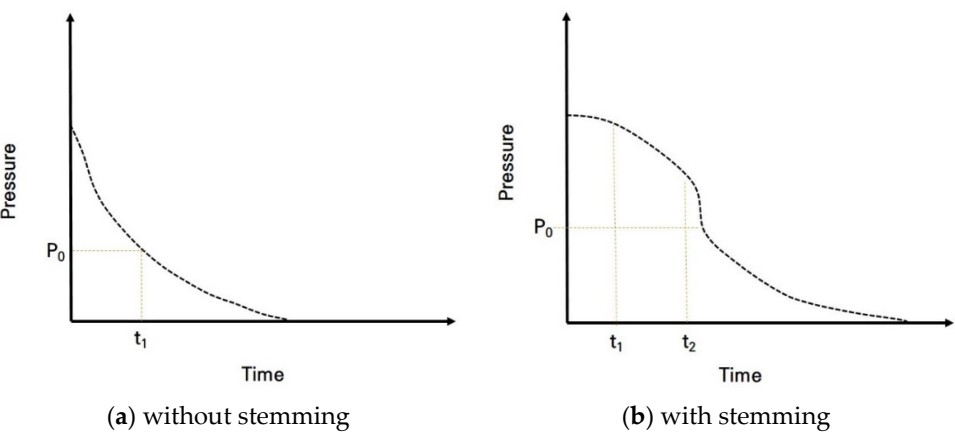

(**a**) without stemming                      (**b**) with stemming

**Figure 16.** Concept curves of pressure and time: (**a**) without stemming, and (**b**) with stemming.

Figure 17 shows the time–pressure results according to the direct measurement of blast hole pressure, where sand- or STF-stemming materials were applied in each experiment. Since the pressure was measured at the upper part of the stemming area, a lower measured peak pressure transmits enough energy to the rock mass around the blast hole and below the stemming area. Stage I indicates detonation durations. When the shock front arrives at the gauge point, the gauge outputs a peak pressure. Stage II indicates the pressure variation as detonation propagates from gauge point to blast hole. Note that during this time, the stemming material begins initial ejection from the blast hole. Finally, in stage III the pressure curve rapidly decreases to atmospheric pressure when the stemming part is completely ejected.

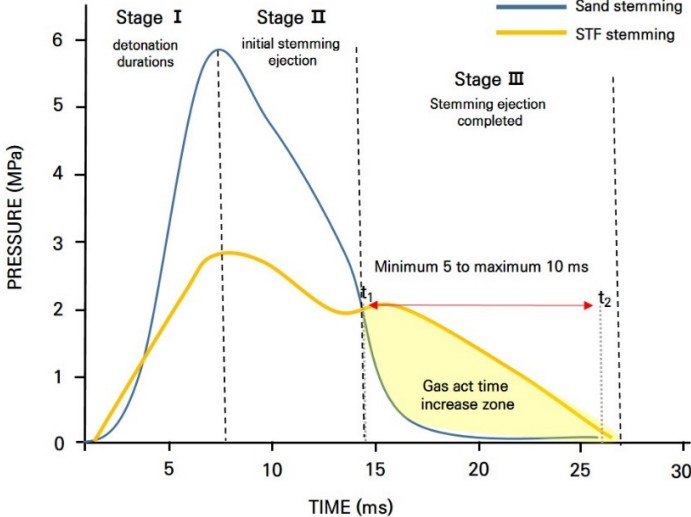

**Figure 17.** The time–pressure curves of the experimental results according to sand or shear thickening fluid (STF) stemming.

In the case of STF-based stemming, the pressure attenuates more slowly than in the sand stemming case. The pressure acts approximately 5–10 ms longer than the blasting gas pressure for the STF-based stemming than for the sand stemming. Eloranta et al. [41] verified that 1 ms of increased gas retention time in a blast hole increased the fragment work done on the rock mass and reduced waste energy. Therefore, an increase in the gas retention time of approximately 5 ms has a significant impact on rock fragmentation.

The purpose of stemming is to increase blasting efficiency by extending the duration of the explosion gas and forming more cracks in the crushing area. Therefore, the ability to resist the gas pressure emitted in the direction of the blast hole inlet implies the stemming performance, which is directly related to the blast efficiency. In this study, after crushing and cracking under the effect of detonation, the pressure applied to the elastic area was measured using a water pressure sensor from a nylon tube inserted into the water tube, and the pressure acting in the elastic area was measured, as shown in Figure 18.

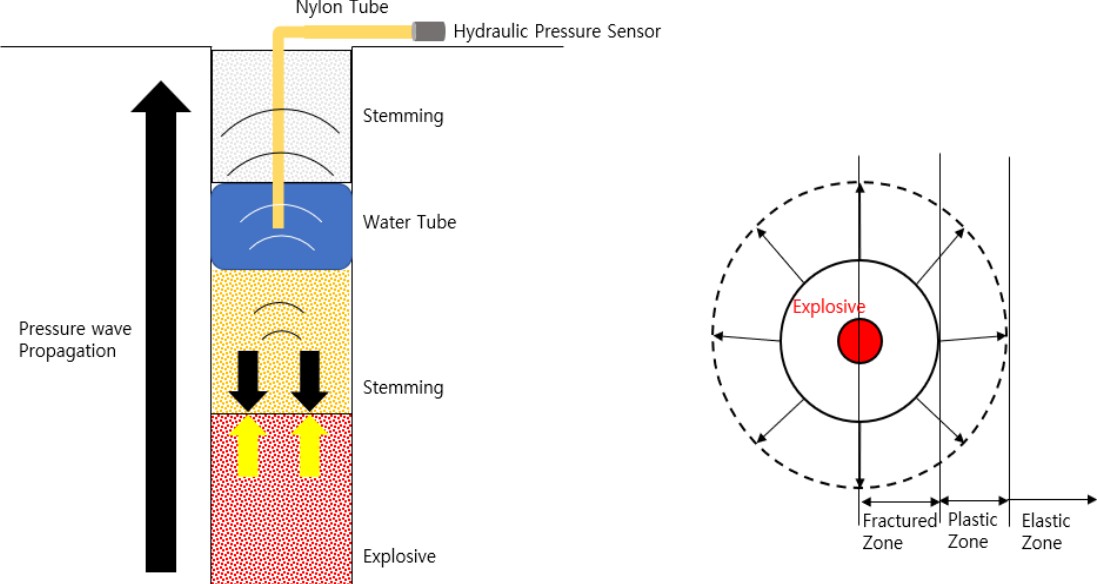

**Figure 18.** Schematic of the measured pressure wave in the blast hole.

Therefore, the pressure behavior in the blast hole owing to the explosive detonation measured through the hydraulic pressure measurement sensor is the pressure value at which the pressure ejected to the upper part of the blast hole by the explosion is attenuated by the stemming material and not by the direct blast pressure. The low peak pressure measured by the hydraulic sensor implies that the stemming material has an excellent pressure constraint against the explosive pressure ejected to the upper part of the blast hole. This suggests that the explosive pressure applied to the lower part of the blast hole is high. In addition, because the explosive pressure delay time measured by the hydraulic pressure sensor is from the trigger to the pressure wave measurement, the longer the delay time from the detonation to the pressure measurement, the more continuous the pressure behavior of the stemming material under the blast hole. Finally, the time at which the explosion pressure returns to atmospheric pressure after the increase indicates the duration of the explosive pressure in the blast hole. This suggests that, the longer the duration of the explosive pressure, the longer the crack propagation time owing to the gas pressure behavior in the blast hole.

The measured pressure behavior in the blast hole correlates with the resistance characteristics of the stemming material. It is judged that the STF-based stemming material will effectively achieve the purpose of the stemming material by resisting the blasting pressure and maintaining relatively high pressure in the blast hole for a longer time compared to the conventional sand stemming material.

## 6. Conclusions

By conducting two blasting experiments, we compared the stemming effects of the developed STF-based stemming material and those of commonly used sand stemming material. The conclusions drawn from the blasting experiments are as follows:

(1) The measured pressure was 2.80 MPa for the STF-based stemming case and 5.84 MPa for the sand stemming case based on the direct dynamic pressure measurement at the top of the blast hole. The lower the measured pressure value, the lower the loss of explosive pressure in the blast hole owing to stemming transfer. In addition, the explosive gas pressure action time in the STF stemming case was 5 ms longer than the sand-stemming case. The longer the duration of the explosive pressure, the greater the energy that can be used to fracture the rock;

(2) The measured pressure behavior in the blast hole correlated with the resistance characteristics of the stemming material. It is judged that the STF-based stemming material can effectively resist the blasting pressure and maintain a relatively high pressure in the blast hole for a longer duration compared to the conventional sand stemming material;

(3) The hydraulic pressure measurement system was developed to measure blasting pressure in a blast hole. This was done by improving the method of measuring the blasting pressure by drilling a dummy hole, which was mainly performed for the explosion pressure measurements. The pressure behavior depending on the voltage of pressure meter with cerabar (PMC) was set through the sensor calibration performed before the blasting experiment, the test method to evaluate the pressure constraint capacity of the stemming material was presented, and the validity of the pressure measurement system in the blasting hole was verified;

(4) As a full-scale bench blasting experiment, a rock pile fragmentation analysis was performed after blasting, and it was confirmed that the average fragment size was reduced by an approximate minimum of 25% to a maximum of 60%, when using STF-based stemming material. This suggests that the secondary crushing work in aggregate quarry mines can be reduced.

**Author Contributions:** Conceptualization, Y.J.; Investigation, C.S.; Methodology, Y.K.; Supervision, S.C.; Writing—original draft, Y.K. All authors have read and agreed to the published version of the manuscript.

**Funding:** This research was supported by a grant from the project "Development of new stemming material and blasting method using shear thickening fluid by high shock-wave reaction", funded by the Korea Institute of Civil Engineering and Building Technology (KICT).

**Institutional Review Board Statement:** Not applicable.

**Informed Consent Statement:** Not applicable.

**Data Availability Statement:** Not applicable.

**Conflicts of Interest:** The authors declare no conflict of interest regarding the publication of this paper.

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
