# Peer review of "Blast Hole Pressure Measurement and a Full-Scale Blasting Experiment in Hard Rock Quarry Mine Using Shock-Reactive Stemming Materials"

_applsci, doi:10.3390/app12178629_

Round 1

Reviewer 1 Report

1 - The title must be improved by adding the type of material (in this case hard rock);

2 - Figure 13, 14, 15, and 16: the fragmented block images: type of software used and user's guide must be added to references;

3 - The sames figures above must be improved and adding (a, b, and c for example);

4 - Introduction must be improved by adding more than 25 references;

5 - Figure 12: Bench blasting results obtained by applying different stemming materials. To compare STF stemming and sand stemming we need to select the same area with the same density of fracturation (natural fracturations such as faults in the rock and fracturation induced by mining explosion. The rock mass of the study area not involved in the paper. According to the histogram mentioned in each figure (13, 14, 15, and 16) we don't see a significant difference between the two methods (STF stemming and Sand stemming).

6 - Figure 11a: Blast hole array and detonator delay time. The detonation diagram is not complete. When we speak about the time of detonation (ms) it is necessary to link the holes (0 to 18) and (1 to 19). Free surface in this case not true. You have to design each case separately.

7 - Discussion and conclusions must be improved.

Author Response

Thank you for your valuable comments, please find my responses as attached below.

Reviewer 2 Report

(1)The introduction part needs to be further simplified. For example, the first three paragraphs of the paper seem to repeat the same meaning.

(2)The introduction part does not explain the background and reason of using STF material as the stemming material in this experiment.

(3)Line 61, "…than that of air [5Under 61 dynamic loading,…" should be  "…than that of air [5]. Under 61 dynamic loading,…"

(4) Some abbreviations should be explained the first time they appear in the text, e.g. STF, PVDF, PMC, PE,..

(5) In formulas (1) and (2) n represents the flow behavior index, while in formula(3)it represents the distribution modulus.

(6) Figure 9, the unit in which the size is given?

(7) The authors do not discuss the innovativeness of the experimental methods and data analysis methods of this paper. For example, is there innovation in the STF-based stemming and the rock pile fragmentation analysis method using image processing?

Author Response

(1)The introduction part needs to be further simplified. For example, the first three paragraphs of the paper seem to repeat the same meaning. 

Response 1 : The introduction has been completely revised in the following order, repeat sentences have been corrected. Thank you for your comments.

1) Energy efficiency improvement of stemming materials

2) Advantages of good stemming material

3) Research trends on fluid-based stemming materials

4) Field test for the evaluation of stemming performance (measurement of gas pressure in the blast hole and evaluation of rock fragment size)

5) Research trends on STF and the background selected as an STF-based stemming material

(2)The introduction part does not explain the background and reason of using STF material as the stemming material in this experiment.

Response 2 : STF is the most current material used to improve the explosion resistance performance in liquid body armors. There is limited research on the application of these STF materials to industrial blasting. The following sentence was added at the end of the introduction. Thank you very much for your valuable comments.

The STF-based stemming material was developed based on the following advantages. 1) STF has excellent sealing properties as it is a fluid-based material. 2) Its viscosity rapidly changes because of external shock, while material compaction or deformation is minimal with respect to the dynamic gas pressure in the blast hole. 3) Using starch as the main base material reduces costs.”

(3)Line 61, "…than that of air [5Under 61 dynamic loading,…" should be  "…than that of air [5]. Under 61 dynamic loading,…"

Response 3 : Thank you for bringing this to our attention. The sentence has been corrected.

(4) Some abbreviations should be explained the first time they appear in the text, e.g. STF, PVDF, PMC, PE,..

Response 4 : The abbreviations have been defined and explained at their first instance of appearance in the manuscript. Thank you.

(5) In formulas (1) and (2) n represents the flow behavior index, while in formula(3)it represents the distribution modulus.

Response 5 : To avoid confusion, the index of n in Equations (3) and (4) has been changed to "a" index. Thank you very much for your insightful comments.

(6) Figure 9, the unit in which the size is given?

Response 6 : Unit in the Blasting Pressure Measurement system schematic are applied in mm and modified by specifying the units in Figure 9.

(7) The authors do not discuss the innovativeness of the experimental methods and data analysis methods of this paper. For example, is there innovation in the STF-based stemming and the rock pile fragmentation analysis method using image processing?

Response 7 : The novelty of this research paper is two-fold. First, a material that instantaneously changes to shock load was developed as a blast stemming material and its performance was verified. The second is that we directly measured the pressure in the upper part of the stemming area inside the blast hole. There limited case studies that directly measure the pressure in the blast hole during blasting. Relevant images and text have been added to the discussion section.

Round 2

Reviewer 2 Report

The author has revised the paper in accordance with the reviewers' comments